# Chewing Behavior Attenuates the Tumor Progression-Enhancing Effects of Psychological Stress in a Breast Cancer Model Mouse

**DOI:** 10.3390/brainsci11040479

**Published:** 2021-04-09

**Authors:** Qian Zhou, Masahisa Katano, Jia-He Zhang, Xiao Liu, Ke-Yong Wang, Mitsuo Iinuma, Kin-ya Kubo, Kagaku Azuma

**Affiliations:** 1Department of Anatomy, School of Medicine, University of Occupational and Environmental Health, Fukuoka 807-8555, Japan; zhou@med.uoeh-u.ac.jp (Q.Z.); jiahezhang@med.uoeh-u.ac.jp (J.-H.Z.); liuxiaoxiao0103@163.com (X.L.); 2Department of Pediatric Dentistry, Asahi University School of Dentistry, Gifu 501-0296, Japan; katano@dent.asahi-u.ac.jp (M.K.); iinuma@dent.asahi-u.ac.jp (M.I.); 3Shared-Use Research Center, School of Medicine, University of Occupational and Environmental Health, Fukuoka 807-8555, Japan; kywang@med.uoeh-u.ac.jp; 4Graduate School of Human Life Science, Nagoya Women’s University, Nagoya 467-8610, Japan; kubo@nagoya-wu.ac.jp

**Keywords:** psychological stress, chewing behavior, breast cancer, glucocorticoid, β2-adrenergic receptor, oxidative stress

## Abstract

We examined whether chewing behavior affects the tumor progression-enhancing impact of psychological stress. Human breast cancer cell line (MDA-MB-231) cells were inoculated into the mammary fat pads of athymic nude mice. The mice were assigned randomly to control, stress, and stress+chewing groups. Psychological stress was created by keeping mice in a transparent restraint cylinder for 45 min, three times a day, for 35 days after cell inoculation. Animals in the stress+chewing group were provided with a wooden stick for chewing on during the psychological stress period. Chewing behavior remarkably inhibited the tumor growth accelerated by the psychological stress. Immunohistochemical and Western blot findings revealed that chewing behavior during psychological stress markedly suppressed tumor angiogenesis and cell proliferation. In addition, chewing behavior decreased serum glucocorticoid levels and expressions of glucocorticoid and β2-adrenergic receptors in tumors. Chewing behavior decreased expressions of inducible nitric oxide synthase and 4-hydroxynonenal, and increased expression of superoxide dismutase 2 in tumors. Our findings suggest that chewing behavior could ameliorate the enhancing effects of psychological stress on the progression of breast cancer, at least partially, through modulating stress hormones and their receptors, and the subsequent signaling pathways involving reactive oxygen and nitrogen species.

## 1. Introduction

Breast cancer is the most commonly occurring cancer in women, affecting more than 1 in 10 women throughout the world, and it has become an increasingly significant global public health problem. Breast cancer is also the major cause of deaths from cancer in women. It is estimated that approximately 15% of all cancer deaths in women are due to breast cancer [1,2]. The development of breast cancer is a multistep process, and the pathogenesis has not yet been fully elucidated. Both individual and environmental factors affect the incidence rate and the prognosis of breast cancer. Family history of breast cancer increases the incidence rate. Those with BRCA1 and BRCA2 mutations have an increased risk of developing breast cancer [3]. Lifestyle factors, including prolonged exposure to estrogens, alcohol use, lack of activity, and obesity, are also considered important risk factors for breast cancer [2]. Recent studies demonstrated that exposure to chronic stressors promotes the initiation, promotion, and progression of breast cancer [4,5].

Long-term psychological stress is increasingly becoming of global health importance because it influences various physiological processes. Exposure to chronic psychological stress induces various pathophysiological alterations, ultimately leading to illness. Chronic psychological stress contributes to the development of a variety of diseases, including cardiovascular disease, diabetes, metabolic syndrome, osteoporosis, and neurodegenerative diseases [6,7,8,9]. In addition, multiple animal and human studies have linked chronic psychological stress with cancer [4,5,10,11,12]. Exposure to chronic psychological stress results in a sustained increase in glucocorticoid and norepinephrine levels via activating the sympathetic nervous system and the hypothalamic–pituitary–adrenal (HPA) axis [13,14,15]. These stress hormones, when binding to their receptors, glucocorticoid receptor (GR) and β2-adrenergic receptor (β2AR), respectively, promote the development, progression, and metastasis of cancers by mediating various signaling pathways [15,16,17,18]. Recent studies have revealed that the prolonged hyperactivity of the HPA axis during the chronic psychological stress response probably alters immune function, leading to the development, growth, and progression of some types of cancer in animal models [4,10,11,12,15].

Activation of either glucocorticoid or norepinephrine receptors can increase the production of reactive oxygen species (ROS) and reactive nitrogen species (RNS) [16,17]. The increase in ROS and RNS in cancer cells stimulates cell proliferation and angiogenesis and drives cancer progression [16,17,19]. 4-Hydroxynonenal (4HNE) is a major oxidative stress factor produced by lipid peroxidation. 4HNE performs an important function in the pathophysiology of different types of cancer and promotes cell proliferation, angiogenesis, and breast cancer invasion [19,20]. Mitochondria are susceptible to oxidative stress owing to their oxygen-dependent metabolism and the presence of redox-sensitive enzymes. Mitochondrial superoxide dismutase (SOD2) is a catalytic antioxidant that protects against spontaneous tumorigenesis caused by oxidative stress [21]. A reduction in SOD2 expression contributes to cell transformation and tumorigenesis by increasing oxidative-stress-induced cancer cell proliferation [21]. Additionally, recent studies showed that stress hormones activate the expression of inducible nitrogen oxide synthase (iNOS), the enzyme that produces nitric oxide (NO), thereby promoting tumor angiogenesis and progression [16,22].

Learning how to effectively manage chronic psychological stress may contribute to slowing the initiation, growth, and metastasis of cancers, and provide a novel strategy for preventing and treating breast cancer. Daily chewing behavior is beneficial for managing stress effectively and preserving physical and mental health. Several animal and human studies revealed that chewing behavior is a useful stress-coping approach [23,24,25,26]. Recent studies demonstrated that chewing behavior under stressful situations can improve stress-associated physical and psychological disorders by inhibiting glucocorticoid and noradrenergic hyperfunction [23,24,27,28,29]. Few systematic studies, however, have examined the impacts of chewing behavior on the development and progression of cancers in an orthotopic cancer microenvironment under stressful conditions. The objective of the present study was therefore to examine the effects of chewing behavior on the growth of breast cancer and to explore the possible underlying molecular mechanisms in athymic nude mice exposed to chronic psychological stress.

## 2. Materials and Methods

### 2.1. Cell Culture

The MDA-MB-231 cell line, human breast adenocarcinoma cells, was obtained from the American Type Culture Collection (Manassas, VA, USA). MDA-MB-231 cells were grown in Dulbecco’s modified Eagle’s medium (DMEM), supplemented with 10% fetal bovine serum, 1% penicillin, and 1% streptomycin. MDA-MB-231 cells were maintained in a humidified incubator containing 5% CO_2_ at 37 °C.

### 2.2. Animal and Xenograft Breast Cancer Model

Female BALB/c athymic nu/nu mice (7 weeks old, n = 27) were obtained from the Japan Charles River Laboratories (Hamamatsu, Shizuoka, Japan) and housed with a standard mouse diet and tap water ad libitum under Specific pathogen-free (SPF) conditions (temperature: 23–25 °C, humidity: 50–60%, light period: 7:00–19:00, dark period: 19:00–7:00). All experimental procedures were carried out under protocols approved by the Ethics Review Committee for Animal Care and Experimentation of the University of Occupational and Environmental Health, Japan (AE 17-023, permission code, 19 January 2018).

Seven days after the animals arrived at our facility, 1 × 10^6^ MDA-MB-231 cells in 10 µL saline were inoculated into the fat pad of the fourth mammary gland, and that day was designated as day 0 after MDA-MB-231 cell inoculation. Afterward, the mice were randomized to the control, stress, or stress +chewing groups (n = 9/group). Mice in the stress and stress+chewing groups were exposed to psychological stress for 45 min, 3 times a day, for 35 days after MDA-MB-231 cell inoculation, as previously reported [14,27]. Briefly, psychological stress was created by keeping the mice in a transparent restraint cylinder (inner diameter: 3.5 cm diameter), in which they could move back and forth, but not turn around. Mice in the stress+chewing group were provided with a wooden stick (diameter: 2 mm) to chew on during the period of restraint stress, as previously described [14,27]. After the experiment was finished, the wooden sticks were examined and all mice demonstrated evidence of chewing behavior. The control mice were neither exposed to restraint stress nor provided with a stick to chew on. The primary tumor size was assessed by using a digital Vernier caliper, and the volume was determined as (length × width^2^)/2.

### 2.3. Tumor Samples Collection and Serum Corticosterone Assay

The blood was extracted from the medial canthus of the mice between 10:00 and 11:00 a.m. on day 35 after the MDA-MB-231 cell inoculation, according to the Institutional Animal Care and Use Committee guidelines, and the serum was separated by centrifugation at 3000 rpm for 10 min at 4 °C. Afterward, all tumors were excised, weighed, and cut in half. One half was fixed in 4% paraformaldehyde solution. The other half was flash-frozen and preserved at −80 °C for Western blot analysis. The serum corticosterone levels were detected using a mouse corticosterone ELISA kit (Assaypro, St. Charles, MO, USA), following the manufacturer instructions.

### 2.4. Immunohistochemistry

Tumor tissues were processed routinely, embedded in paraffin, and sections of 5 μM thickness were prepared. After deparaffinization and rehydration, the tissue sections were treated with 10% H_2_O_2_ for 10 min for inhibition of endogenous peroxidase activity. After washing in PBS, sections were incubated with Protein Block, Serum Free (Dako, Tokyo, Japan) for 15 min to reduce nonspecific binding. For immunohistochemical staining, sections were first incubated with anti-CD31 antibody (1:300, Abcam, Cambridge, MA, USA), anti Ki67 antibody (1:200, Abcam), anti-GR antibody (1:400, Cell Signaling Technology, Danvers, MA, USA), or anti-β2AR antibody (1:50, Abcam), and then treated with biotinylated goat anti-rabbit IgG and streptavidin peroxidase complex (Nichirei Biosciences Inc., Tokyo, Japan) for 30 min. They were stained with 3,3′-diaminobenzidine tetrahydrochloride (Nichirei Biosciences Inc., Tokyo, Japan) and then counterstained with hematoxylin.

The slides were examined using a light microscope (Olympus, BX50, Tokyo, Japan) linked to a digital camera. To calculate the percentage of the CD31-positive microvessel area, a proliferation marker (Ki67-), glucocorticoid receptor (GR-), and beta-2 adrenergic receptor (β2AR)-positive cells, 9 fields per section, at 400× were randomly selected for evaluation. The immunohistochemical images were evaluated using ImageJ (NIH, Bethesda, MD, USA), as described previously [30,31,32,33].

### 2.5. Western Blot Analyses

Total protein extraction from tumor tissues was performed using a modified radioimmunoprecipitation assay buffer (Millipore, Burlington, MA, USA), and the protein concentration was measured by using a bicinchoninic acid (BCA) protein assay kit (Thermo Scientific, Waltham, MA, USA). We applied odium dodecyl sulfate-polyacrylamide gel electrophoresis (Invitrogen, Carlsbad, CA, USA) to separate proteins (30 µg) with sodium dodecyl sulfate-polyacrylamide gel electrophoresis (Invitrogen, Carlsbad, CA, USA) and blotted onto polyvinylidene difluoride membranes (Millipore, Burlington, MA, USA). After the membrane was blocked, immunoblotting was carried out by using the following antibodies at 4 °C 24 h: anti-GAPDH (1:1000, Cell Signaling Technology), anti-CD31 (1:500, #ab28364, Abcam, Cambridge, MA, USA), anti-Ki67 (1:500, #ab16667, Abcam, Cambridge, MA, USA), anti-GR (1:1000, Cell Signaling Technology), anti-β2AR (1:100, Abcam), anti-iNOS (1:2000, Thermo Scientific), anti-SOD2 (1:1000, Cell Signaling Technology), and anti-4HNE (100 µg/mL, Japan Institute for the Control of Aging, Fukuori, Shizuoka, Japan). We incubated the immunoblotting membranes with a secondary antibody (1:1000, Cell Signaling Technology) for 1 h, and then visualized with the use of an ECL kit (GE Healthcare Bio-Science, Chicago, IL, USA). The target protein bands were determined with the aid of an Ez-Capture MG System (Atto Corporation, Tokyo, Japan), and the densitometric analysis of the Western blot band was carried out according to the Scion Image software program (version 4.0.2; Scion Corp., Frederick, MD, USA). In the Western blot analyses, GAPDH antibody was used as an internal loading control protein to normalize the levels of CD31, Ki67, GR, β2AR, iNOS, and 4HNE. All primary antibodies used in this study are shown in Table 1.

### 2.6. Statistical Analysis

All values are expressed as mean ± Standard Error of Mean (SEM). Statistical analyses were carried out by using SPSS version 22. Multiple comparisons were performed using 1-way analysis of variance (ANOVA) followed by Tukey–Kramer’s post hoc multiple comparison tests. Differences were considered statistically significant at *p* < 0.05.

## 3. Results

### 3.1. Chewing Behavior, Psychological Stress, and Tumor Growth

Animal body weights were monitored during the experimental period. Body weight tended to be lower in the stress group, but we did not find any significant differences regarding body weight among the three groups (*p* > 0.05, Figure 1A). At 7, 14, 21, and 35 days after tumor cell inoculation, the tumor volume in the stress group was significantly larger than that in the control and stress+chewing groups (*p* < 0.01, Figure 1B). The volume of the dissected tumor in the stress group was visually larger than in the control group, and the tumor volume in the stress+chewing group appeared smaller than that in the stress group (Figure 1C). The tumor weight in the stress group was also significantly higher than that in the control group (*p* < 0.01), and the tumor weight in the stress+chewing group was remarkably lower than that in the stress group (*p* < 0.01, Figure 1D).

### 3.2. Chewing Behavior, Psychological Stress, and Tumor Angiogenesis and Proliferation

Tumor angiogenesis was measured by detecting the expression of CD31 (vessel endothelial cell marker). Tumor cell proliferation was determined by analyzing the expression of Ki67 (cancer cell proliferation marker). Representative photomicrographs of CD31-positive cells and Ki67-positive cells in the tumor are shown in Figure 2A. The microvessel density in the stress+chewing group was significantly lower than that in the stress group (*p* < 0.01, Figure 2B), and the protein expression of CD31 in the stress+chewing group was significantly lower than that in the stress group (*p* < 0.01, Figure 2D), indicating that chewing behavior during psychological stress remarkably attenuated the increased tumor vascularity. The percentage of Ki67-positive tumor cells in the stress group was markedly higher than that of the control and stress+chewing groups (*p* < 0.01, Figure 2C), and the protein expression of Ki67 in the stress+chewing group was significantly lower than that of the stress group (*p* < 0.01, Figure 2E), suggesting that chewing behavior during psychological stress remarkably reduced the tumor cell proliferation.

### 3.3. Chewing Behavior, Psychological Stress, Serum Corticosterone Levels, and Glucocorticoid Receptor Expression in the Tumor

The serum corticosterone levels in the stress group were significantly higher than in the control and stress+chewing groups (*p* < 0.01, Figure 3A). Compared with the control group, chronic restraint stress caused a significant increase in the protein expression of GR in the tumor (*p* < 0.01), and chewing behavior during psychological stress significantly decreased the protein expression of GR in the tumor (*p* < 0.01, Figure 3B). Immunohistochemical staining showed a higher percentage of GR-positive tumor cells in the stress group in comparison with the control and stress+chewing groups (*p* < 0.01, Figure 3C,D).

### 3.4. Chewing Behavior, Psychological Stress, and β2-Adrenergic Receptor Expression in the Tumor

The immunohistochemical images of β2AR-positive cells in the tumor are shown in Figure 4A. The relative area of β2AR-positive tumor cells in the stress group was markedly higher than in the control group (*p* < 0.01), and the relative area of β2AR-positive tumor cells in the stress + chewing group was significantly lower than in the stress group (*p* < 0.01, Figure 4B). Western blot analysis showed that psychological stress induced a significant increase in the expression level of β2AR protein in the tumor compared with the control group (*p* < 0.01). Chewing behavior during psychological stress significantly decreased the expression level of β2AR in the tumor in comparison with the stress group (*p* < 0.01, Figure 4C).

### 3.5. Chewing Behavior, Psychological Stress, and Oxidative Stress in the Tumor

Oxidative stress was evaluated by detecting the expression of iNOS (a family of enzymes catalyzing the production of nitric oxide), 4HNE (one of several lipid peroxidation end-products), and SOD2 (mitochondrial superoxide scavenger) in the tumor. Chronic restraint stress led to an elevation in the expression level of iNOS and 4HNE, and a significant reduction in the protein expression of SOD2, and chewing during the chronic restraint stress caused reduction in the protein expression of iNOS and 4HNE, and elevation in the protein expression of SOD2 in comparison with the stress group (*p* < 0.01, Figure 5), suggesting that chewing behavior during psychological stress reduced oxidative stress in the tumor.

## 4. Discussion

The key findings of this study revealed that chewing behavior during psychological stress attenuated the enhancing effect of psychological stress on tumor progression by alleviating tumor angiogenesis and cell proliferation in a mouse model of breast cancer. Angiogenesis is a necessary process that enables tumor growth and metastasis by providing oxygen and nutrients [34,35]. Consistent with previous reports, psychological stress accelerated tumor angiogenesis by promoting excessive vascularity and cell proliferation in the tumor [5,10,11,36]. Our data indicated that chewing on a wooden stick under psychologically stressful conditions markedly attenuated the excessive vascularity of the tumor and cancer cell proliferation. Glucocorticoid and norepinephrine levels are increased in individuals with acute or chronic psychologic stress [4,10,11,13,37]. Glucocorticoid, as a type of stress hormone, binds to GR and plays crucial roles in numerous biologic processes, including cell proliferation, apoptosis, metabolism, immune, and inflammatory reactions [15,38]. An ever-increasing body of studies has confirmed that abnormally elevated glucocorticoid levels are associated with faster tumor progression, particularly in breast cancer [15,39,40]. Persistently higher expression of GR has a causal association with enhanced angiogenesis and cancer cell proliferation [15,41,42]. Chewing, representing a useful approach for coping with stress, could ameliorate stress-related physical and mental diseases, by suppressing the hyperactivity of the HPA axis through reducing circulating glucocorticoid levels [23,43,44]. In the present study, we found that allowing mice to chew on a stick during psychological stress markedly reduced the serum corticosterone levels and decreased the expression of GR in tumor cells in comparison with mice exposed to psychological stress. We consider that chewing behavior could attenuate the steep glucocorticoid increase induced by stress and inhibit tumor progression mediated by downregulating GR expression.

Previous studies revealed that chronic psychological stress drives tumorigenesis, tumor progression, and metastasis, which are mediated via bidirectional signaling among tumor cells and their surrounding microenvironment [45]. β-adrenergic signaling is linked with the stress response [46,47]. Intracellular β2AR signaling is involved in the modulation of tumorigenesis and tumor progression [46,48], and has been identified as a target to regulate cancer progression [18,46], especially in breast cancer and gastric cancer [49,50,51]. Recent studies indicated that β2AR is highly expressed in many cancer tissues, and enhances tumorigenesis, angiogenesis, and metastasis in breast cancer and gastric cancer under chronic stress [13,51,52]. Here, we revealed that chronic psychological stress markedly increased the expression of β2AR in breast cancer cells. The increased β2AR expression was significantly decreased by chewing behavior during psychological stress. Consequently, we consider that chewing behavior during chronic psychological stress downregulates β2AR expression, thereby suppressing effects of chronic stress on tumor development and progression.

Accumulating evidence indicates roles of oxidative stress in the pathogenic mechanism and progression of a variety of cancers [19,53,54,55]. Chronic psychological stress induces a significant increase in oxidative stress, potentially affecting tumorigenesis and progression, which may be attributed to sustained increases in glucocorticoid levels [17,53,55]. Stress hormones could induce DNA damage, potentially affecting tumorigenesis, through the overproduction of ROS and RNS [17,56]. Animal studies demonstrate activating β2AR induces oxidative stress by activating phosphorylation of protein kinase A, producing ROS, and producing oxidative phosphorylation in the cells [13,55,57]. Furthermore, iNOS can trigger the cellular injury by producing NO [16,58]. Overexpression of iNOS is related to enhanced tumorigenesis, tumor angiogenesis, and proliferation by increasing oxidative stress [17,59,60]. Given that oxidative stress is a major contributor to chronic restraint-stress-induced tumorigenesis and tumor progression, we explored the effects of chewing on chronic restraint-stress-induced oxidative stress in breast cancer. Our results demonstrated that chronic restraint stress caused a prominent increase in the expression of iNOS and 4HNE proteins, and a decrease in the expression of SOD2 protein, indicating that chronic restraint stress enhanced angiogenesis and cell proliferation of the breast cancer tissue via the generation of oxidative stress. Simultaneously, chewing during restraint stress clearly reduced oxidative stress by downregulating the expression of iNOS and 4HNE, and upregulating the expression of SOD2. Taken together, our findings demonstrate that chewing behavior during psychological stress attenuates the enhancing effects of chronic stress on breast cancer development, mediated by ROS and RNS activity.

Recent clinical studies showed that several healthy lifestyle and behavioral factors, including physical activity, diet, and meditation, could enhance coping capacity, lead to beneficial effects on immune function, and improve breast cancer survival rates [61,62]. Chewing gum ameliorates the stress response, relieves negative moods, and decreases cortisol levels during various stressful conditions in humans [23,24,26]. We consider that chewing gum may therefore be a useful practical behavior to reduce or delay the development of breast cancer and to improve the impacts of stress on the progression of the breast cancer.

Our study has several limitations. We investigated the impacts of chewing behavior on chronic psychological stress-associated breast cancer progression. It is also necessary to investigate whether chewing behavior induces any changes in stress-free chewing animals. Mice can chew a stick under restraint stress conditions, but they do not chew continuously while freely moving around their home cage. Consequently, it is hard to perform such experiments with stress-free chewing animals. Another limitation of the study is that we did not adequately investigate the intracellular signaling pathways. Cancer angiogenesis and cell proliferation induced by oxidative stress involve several intracellular signaling pathways, including nuclear factor κB, hypoxia-inducible factors, and vascular endothelial growth factor [19,53,59]. Further studies are necessary to explore the potential molecular mechanisms of the ameliorative effects of chewing behavior on chronic psychologic-stress-associated tumor progression.

## 5. Conclusions

The findings of the present study revealed that chewing behavior ameliorates the enhancing effects of chronic psychological stress on breast cancer progression in a xenograft mouse model, at least partially, by modulating stress hormones and their receptors, and the subsequent signaling pathways of reactive oxygen and nitrogen species.

## Figures and Tables

**Figure 1 brainsci-11-00479-f001:**
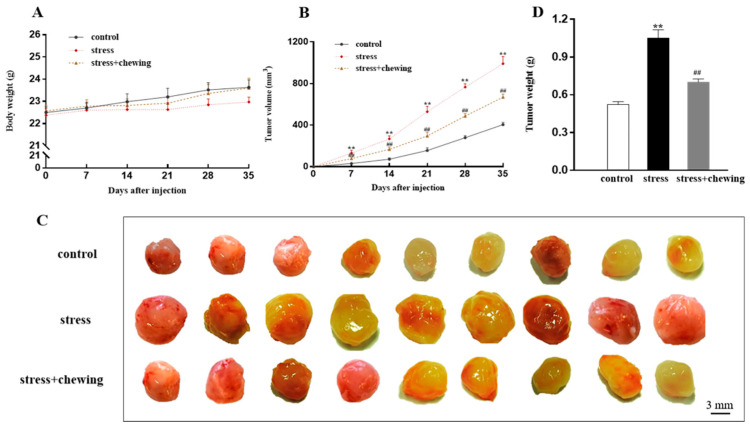
Chewing behavior during psychological stress inhibited tumor growth. (**A**) Mouse body weight growth curves. (**B**) Tumor growth curves. (**C**) Representative photograph of tumors after dissection. (**D**) Tumor weight on day 35 after inoculation. (** *p* < 0.01 vs. control group, ## *p* < 0.01 vs. stress group, n = 9/group). All data are expressed as mean ± SEM.

**Figure 2 brainsci-11-00479-f002:**
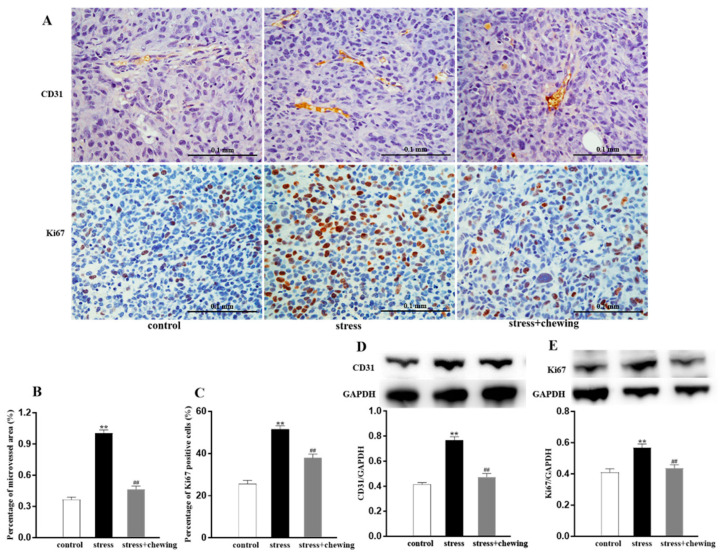
Chewing behavior during psychological stress alleviated tumor angiogenesis and proliferation. (**A**) Representative photographs of CD31-positive cells and Ki67-positive cells in the tumor. (**B**) The percentage of microvessel area in the tumor. (**C**) The percentage of Ki67-positive cells in the tumor. (**D**) The expression level of CD31 in the tumor. (**E**) The expression level of Ki67 in the tumor. (** *p*< 0.01 vs. control group, ## *p* < 0.01 vs. stress group, n = 9/group). All data are expressed as mean ± SEM.

**Figure 3 brainsci-11-00479-f003:**
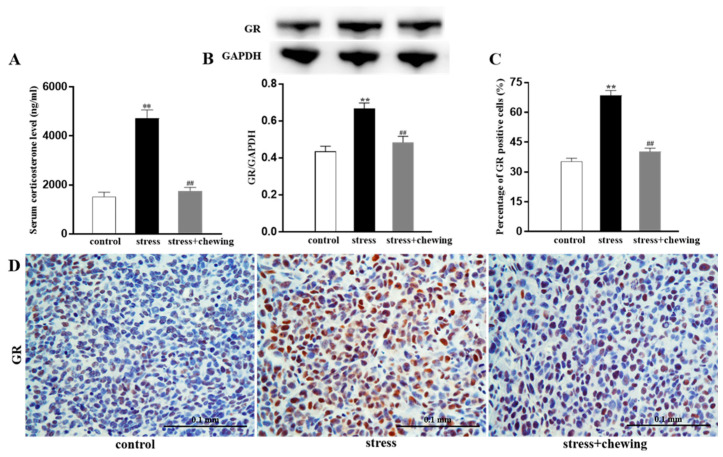
Chewing behavior during psychological stress decreased serum glucocorticoid levels and expression of GR in the tumor. (**A**) Serum glucocorticoid levels. (**B**) The expression level of GR protein in the tumor. (**C**) Percentage of GR-positive cells in the tumor. (**D**) Representative photographs of GR-positive cells in the tumor. (** *p* < 0.01 vs. control group, ## *p* < 0.01 vs. stress group, n = 9/group). All data are expressed as mean ± SEM.

**Figure 4 brainsci-11-00479-f004:**
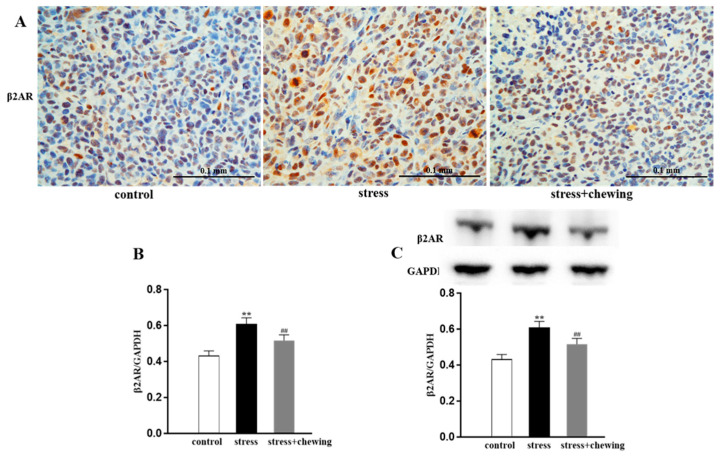
Chewing behavior during psychological stress decreased the expression of β2AR in the tumor. (**A**) Representative photographs of β2AR-positive cells in the tumor. (**B**) The percentage of β2AR-positive cells in the tumor. (**C**) The expression level of β2AR protein in the tumor. (** *p* < 0.01 vs. control group, ## *p* < 0.01 vs. stress group, n = 9/group). All data are expressed as mean ± SEM.

**Figure 5 brainsci-11-00479-f005:**
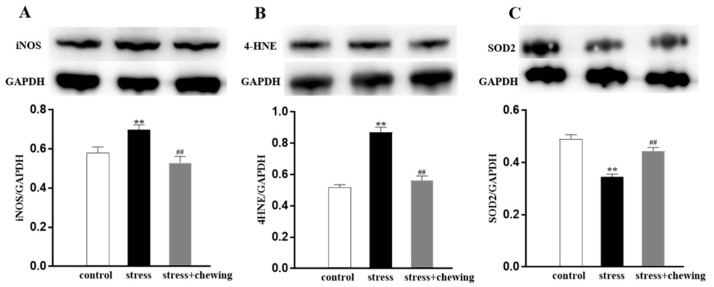
Chewing behavior during psychological stress improved oxidative stress in the tumor. (**A**) The expression level of iNOS protein in the tumor. (**B**) The expression level of 4HNE protein in the tumor. (**C**) The expression level of SOD2 protein in the tumor (** *p* < 0.01 vs. control group, ^##^
*p* < 0.01 vs. stress group, n = 9/group). All data are expressed as mean ± SEM.

**Table 1 brainsci-11-00479-t001:** Antibodies used for Western blot and immunohistochemistry.

Antibody	Manufacturer	Product Number	Dilution (Western Bolt (WB))	Dilution (Immunohistochemistry (IH))
GAPDH	Cell Signaling Technology	2118	1:1000	−
CD31	Abcam	ab28364	1:500	1:300
Ki67	Abcam	ab16667	1:1000	1:200
GR	Cell Signaling Technology	12041	1:1000	1:400
β2AR	Abcam	ab135641	1:100	1:50
iNOS	Thermo Scientific	PA3-030A	1:2000	−
SOD2	Cell Signaling Technology	13141	1:1000	−
4HNE	Japan Institute for the Control of Aging	MHN-100P	100 µg/mL	−

## Data Availability

Not applicable.

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
