# Peer review of "Chewing Behavior Attenuates the Tumor Progression-Enhancing Effects of Psychological Stress in a Breast Cancer Model Mouse"

_brainsci, 2021, doi:10.3390/brainsci11040479_

Round 1

Reviewer 1 Report

The authors sufficiently addressed the raised critical points. The overall manuscript is suitable for publication. However, I did not have access to the supplementary data with only the main manuscript is available for review in the submission system. It might not be an author's issue but rather a submission system. 

Author Response

We thank the reviewer for the positive comments on our manuscript.

The supplementary data includes the original western blot images.

Reviewer 2 Report

The authors investigated the effects of chewing behavior on breast cancer growth in mice exposed to chronic psychologic stress, my recommendation is accepted after minor revisions.

The subtitles of the results highlight what should be concluded from observing the results obtained, so they should be changed as follows

  1. Results 178

3.1. Chewing behavior, psychologic stress, and tumor growth

3.2. Chewing behavior, psychologic stress, and tumor angiogenesis and proliferation

3.3. Chewing behavior, psychologic stress, serum corticosterone levels, and glucocorticoid receptor expression in the tumor

3.4. Chewing behavior, psychologic stress, and β2-adrenergic receptor expression in the tumor

3.5. Chewing behavior, psychologic stress, and oxidative stress in the tumor.

The title of figures from 1 to 5 can remain unchanged

Author Response

We appreciate this reviewer’s thoughtful comments.

We revised the subtitles from 3.1 to 3.5 according to the reviewer’s suggestion.

Reviewer 3 Report

End of Discussion: Do we have any information on substances or molecules released upon chewing behaviour from the wood stick ? Any information on potential biological action of supposed substances ?

It would be interesting to investigate the potential effects of adding biologically-acting substances to the wood. This would possibly add to to the chewing action per se on hormone release.

Author Response

We thank this reviewer for the constructive comments.

We have no information available on substances or molecules released upon chewing behavior from the wood stick. The chewing behavior could ameliorate psychologic stress responses partially through regulating stress hormones and their receptors.

The wooden stick, we used in this study, is made from virgin white birth tree.

It is interesting to investigate the potential effects of adding biologically-acting substances to the wood, as you mentioned. We consider to do such study in the near future, according to your suggestion.

This manuscript is a resubmission of an earlier submission. The following is a list of the peer review reports and author responses from that submission.

Round 1

Reviewer 1 Report

This manuscript investigated the effects of chewing on the breast cancer cell growth in nude mice exposed to chronic restraint stress. The authors inoculated MDA-MB-231 cells into mammary fat fad of immunodeficiency nude mice and divided into control, stress, and stress+chewing groups. The findings in this manuscript suggest that chewing could ameliorate the enhancing effect of chronic stress on breast cancer progression, at least partially through modulating stress hormones and their receptors, and the subsequent signaling pathways of reactive oxygen species and reactive nitrogen species. However, the authors didn’t provide enough data to investigate the how chewing releasing stress induced hormones regulation, ROS, NO and SOD2 productions. Instead, the findings (figures) present in this manuscript are very superficial. An integration systemic analysis of alternations on cytokine in mouse serum and gene expression in xenograft should be detail investigated. Some critical and major commands are listed below.

1: An cytokine ELISA array is absolutely needed in this manuscript to reveal the cytokine alternation in mouse serum that affects breast cancer growth in this unique mouse model.

2: Figure 1C showed only 3 xenograft tumors in each mouse group. According to the material and method section, there were should be 9 mice in each group. The figure1C should illustrated all xenograft tumors.

3: Aa appropriate positive control is missing in this mouse model. Since this manuscript investigate the stress affects tumor progression, GR antagonist (RU486 or mifepristone) is suggested to include in this investigation.

4: There are many genes in cells would be affected by stress, such as NF-KB and STAT-3 activation, IL6, IL8, VEGF, VEGFR, TNF-a, PAI-1, MMP2, MMP9, MCP-1 and inflammasome genes induction. Therefore, an RNA-seq information should be done to investigate the whole gene alternation among these xenograft tumors.

5: A parental MDA-MB-231 cell-based experiments should be also accessed in this manuscript, such as cell proliferation, invasion and invasion assay.

Author Response

Responses to the comments of Reviewer 1

Thank you for your valuable comments. The response to your comments is as follows.

This manuscript investigated the effects of chewing on the breast cancer cell growth in nude mice exposed to chronic restraint stress. The authors inoculated MDA-MB-231 cells into mammary fat fad of immunodeficiency nude mice and divided into control, stress, and stress+chewing groups. The findings in this manuscript suggest that chewing could ameliorate the enhancing effect of chronic stress on breast cancer progression, at least partially through modulating stress hormones and their receptors, and the subsequent signaling pathways of reactive oxygen species and reactive nitrogen species. However, the authors didn’t provide enough data to investigate the how chewing releasing stress induced hormones regulation, ROS, NO and SOD2 productions. Instead, the findings (figures) present in this manuscript are very superficial. An integration systemic analysis of alternations on cytokine in mouse serum and gene expression in xenograft should be detail investigated. Some critical and major commands are listed below.

1: An cytokine ELISA array is absolutely needed in this manuscript to reveal the cytokine alternation in mouse serum that affects breast cancer growth in this unique mouse model.

Response: As you indicated, it is necessary to detect the relative levels of various cytokines in mouse serum and tissue by using cytokine array kit, in order to understand the detailed mechanisms. According to the editor’s suggestion, we have to upload the revised file within 10 days. Therefore, we consider to determine the cytokine levels in our next study.

2: Figure 1C showed only 3 xenograft tumors in each mouse group. According to the material and method section, there were should be 9 mice in each group. The figure1C should illustrated all xenograft tumors.

Response: Thank you for your indication. We revised Figure 1C as you suggested.

3: An appropriate positive control is missing in this mouse model. Since this manuscript investigate the stress affects tumor progression, GR antagonist (RU486 or mifepristone) is suggested to include in this investigation.

Response: A recent study (Obradović et al. Nature 2019, 567:540-544) showed that treatment of MDA-MB 231 cells with GR antagonist mifepristone had no effect on tumor volume, but increased cancer cell displacement. We would like to examine tumor progression after treatment with GR antagonist in the near future.

4: There are many genes in cells would be affected by stress, such as NF-KB and STAT-3 activation, IL6, IL8, VEGF, VEGFR, TNF-a, PAI-1, MMP2, MMP9, MCP-1 and inflammasome genes induction. Therefore, an RNA-seq information should be done to investigate the whole gene alternation among these xenograft tumors.

Response: Thank you for your indication. The stress and chewing might cause the alternation of the expression of the above genes. According to the editor’s suggestion, we have to upload the revised file within 10 days. Therefore, we consider to determine the above gene alternation in our next study.

5: A parental MDA-MB-231 cell-based experiments should be also accessed in this manuscript, such as cell proliferation, invasion and invasion assay.

Response: As you indicated, it is better to examine the MDA-MB-231 cell-based cell proliferation, invasion and invasion assays. Due to time constraints, we would like to do these experiments in our next study.

Reviewer 2 Report

Manuscript “Chewing attenuates the tumor progression-enhancing effect of chronic stress in a mouse model of breast cancer” by Zhou et al. is a well-designed and well-written in vivo study on relationships between stress-relieving behavior and tumor progression. Authors present compelling evidence on the mechanistic relationships between glucocorticoid elevation and tumorogenesis. The study limitations are well identified and stated.

Minor criticism

  1. Materials and methods lacking antibody product numbers with multiple antibodies for the target protein available from the same vendors the exact product numbers are crucial for the study reproducibility. Authors present multiple antibodies in immunostaining and WB experiments. Arranging the antibody information into the table would make materials and methods easier to follow.
  2. Figure 2 – authors supported all other immunohistochemistry findings with western data but not Ki67 or CD31. If authors have lysates from tumors remaining, it will strengthen the figure 2.
  3. Discussion – authors should include the clinical implication of their findings for the human patients in the discussion. Currently, the discussion only focused on mice.
  4. “DA-MB-231 cell line, a human breast adenocarcinoma cell line” – please include the image of the DA-MB-231 cell culture as supplementary data.
  5. Supplementary Figures – Pars of the blot are cut/obscured (i.e., Figure S8); please present whole uncut images. Please make sure that all supplementary membranes are presented in the way they were detected. Also, mark the areas presented in the manuscript. If some bands are present in the membranes that were not presented in the manuscript, please label them and describe why the band was discarded. The authors do not present the ladder although they used it; the ladder is partially visible in the few full membranes. Authors gel documentation system does not support taking visible-light images to include the ladder? Please state the ladder used in the materials and methods as the mains of resolving the band size.

More of the missed opportunities rather than criticism.

  1. Authors could also study the effects of metastasis of the cancer cells; since the inoculated cells are human, they can be detected in the mouse tissues by rabbit or goat anti-human secondary antibody either by immunohistochemistry or by western blot analysis from the lymph nodes, lungs, and liver with tumors serving as a positive control for the staining. The authors could confirm the tumor content as a human cells with the existing histological material without additional animal experiments.
  2. Injecting the tumors with DHE staining and producing the cryosections would provide more direct evidence for the oxidative stress than 4HNE. If authors still have flash-frozen unfixed tumors, the cryosections and the DHE staining can be performed without additional experiments to further demonstrate oxidative stress in the tumors.
  3. Also, if stress-relieving behavior is associated with the glucocorticoids, exogenous administration of the glucocorticoids through drinking water (injection would induce the stress response) should aggravate the tumorogenesis similar to the stress-induced response. Perhaps authors can use this approach in their subsequent studies.

Author Response

Responses to the comments of Reviewer 2

Minor criticism

  1. Materials and methods lacking antibody product numbers with multiple antibodies for the target protein available from the same vendors the exact product numbers are crucial for the study reproducibility. Authors present multiple antibodies in immunostaining and WB experiments. Arranging the antibody information into the table would make materials and methods easier to follow.

Response: We provided the product numbers of all antibodies used for immunohistochemistry and western blot in this study by attachment (Table 1).

  1. Figure 2 – authors supported all other immunohistochemistry findings with western data but not Ki67 or CD31. If authors have lysates from tumors remaining, it will strengthen the figure 2.

Response: According to your suggestion, we added western blot findings of CD31 and Ki67 in Figure 2.

  1. Discussion – authors should include the clinical implication of their findings for the human patients in the discussion. Currently, the discussion only focused on mice.

Response: We added one paragraph for human patients in the discussion, according to your suggestion as follows. “Recent clinical studies indicated that several healthy lifestyle and behavioral factors, including physical activity, diet and meditation could enhance coping capacity, lead to beneficial effects on immune function, and improve breast cancer survival rates [61, 62]. Chewing gum ameliorates the stress response, alleviates a negative mood, and reduces cortisol levels during various psychologic stress in human [23, 24, 26]. We consider that chewing gum may thus be a simple method to attenuate or delay the development of breast cancer and to ameliorate the effects of stress on the progression of the breast cancer. ”.

  1. “MDA-MB-231 cell line, a human breast adenocarcinoma cell line” – please include the image of the MDA-MB-231 cell culture as supplementary data.

Response: We added the image of the MDA-MB-231 cell culture as supplementary data (S10).

  1. Supplementary Figures – Pars of the blot are cut/obscured (i.e., Figure S8); please present whole uncut images. Please make sure that all supplementary membranes are presented in the way they were detected. Also, mark the areas presented in the manuscript. If some bands are present in the membranes that were not presented in the manuscript, please label them and describe why the band was discarded. The authors do not present the ladder although they used it; the ladder is partially visible in the few full membranes. Authors gel documentation system does not support taking visible-light images to include the ladder? Please state the ladder used in the materials and methods as the mains of resolving the band size.

Response: According to your suggestion, we presented whole uncut western blot images (S1-S9).

More of the missed opportunities rather than criticism.

  1. Authors could also study the effects of metastasis of the cancer cells; since the inoculated cells are human, they can detected in the mouse tissues by rabbit or goat anti-human secondary antibody either by immunohistochemistry or by western blot analysis from the lymph nodes, lungs, and liver with tumors serving as a positive control for the staining. The authors could confirm the tumor content as a human cells with the existing histological material without additional animal experiments.

Response: Thank you for your valuable comments. We found there were several metastases in the lungs, livers and ovaries. We consider to prepare all metastatic findings as a separate manuscript.

  1. Injecting the tumors with DHE staining and producing the cryosections would provide more direct evidence for the oxidative stress than 4HNE. If authors still have flash-frozen unfixed tumors, the cryosections and the DHE staining can be performed without additional experiments to further demonstrate oxidative stress in the tumors.

Response:

It is a good method to detect the generation of ROS using Dihydroethidium (DHE) assay kit, as a fluorescent probe. Unfortunately, all tissues were used for analyzing immunohistochemistry and molecular biology. We consider to examine ROS generation using DHE method in our next study.

  1. Also, if stress-relieving behavior is associated with the glucocorticoids, exogenous administration of the glucocorticoids through drinking water (injection would induce the stress response) should aggravate the tumorogenesis similar to the stress-induced response. Perhaps authors can use this approach in their subsequent studies.

Response: Thank you for your suggestion. The effect of exogenous administration of the glucocorticoids on breast cancer progression and metastasis was reported recently (Obradović et al. Nature 2019, 567:540-544). We would like to compare the effects of restraint stress and exogenous administration of the glucocorticoid on breast cancer progression in our subsequent study.
